# Clinical Practice Guidelines for Diagnosis, Treatment and Follow-Up of Exocrine Pancreatic Ductal Adenocarcinoma: Evidence Evaluation and Recommendations by the Italian Association of Medical Oncology (AIOM)

**DOI:** 10.3390/cancers12061681

**Published:** 2020-06-24

**Authors:** Nicola Silvestris, Oronzo Brunetti, Alessandro Bittoni, Ivana Cataldo, Domenico Corsi, Stefano Crippa, Mirko D’Onofrio, Michele Fiore, Elisa Giommoni, Michele Milella, Raffaele Pezzilli, Enrico Vasile, Michele Reni

**Affiliations:** 1Medical Oncology Unit–IRCCS IstitutoTumori “Giovanni Paolo II” of Bari, 70124 Bari, Italy; n.silvestris@oncologico.bari.it (N.S.); dr.oronzo.brunetti@tiscali.it (O.B.); 2Department of Biomedical Sciences and Human Oncology-University of Bari Medical School, 70124 Bari, Italy; 3Oncology Clinic, AOU Ospedali Riuniti, Polytechnic University of Marche, 60121 Ancona, Italy; alebitto@tiscali.it; 4Department of Pathology, Hospital Cà Foncello of Treviso, 31100 Treviso, Italy; cataldo.ivana@gmail.com; 5Medical Oncology Unit Azienda Ospedaliera San Giovanni Calibita Fatebene fratelli Roma, 00186 Roma, Italy; domenico.corsi@libero.it; 6Division of Pancreatic Surgery, Vita-Salute University, San Raffaele Scientific Institute, 20132 Milan, Italy; crippa1.stefano@hsr.it; 7Department of Radiology, G. B. Rossi University Hospital, University of Verona, 37129 Verona, Italy; mirko.donofrio@univr.it; 8Radiation Oncology, Campus Bio-Medico University, 00128 Rome, Italy; m.fiore@unicampus.it; 9Medical Oncology Unit, Department of Oncology and Robotic Surgery, AOU Careggi, 50139 Florence, Italy; elisa.giommoni@gmail.com; 10Section of Medical Oncology, Department of Medicine, University of Verona and University Hospital Trust, 37129 Verona, Italy; michele.milella@aovr.veneto.it; 11Department of Gastroenterology, San Carlo Hospital, 85100 Potenza, Italy; raffaele.pezzilli@gmail.com; 12Division of Medical Oncology, Pisa University Hospital, 56124 Pisa, Italy; envasile@tin.it; 13Department of Medical Oncology, IRCCS Ospedale San Raffaele, 20132 Milan, Italy

**Keywords:** pancreatic ductal adenocarcinoma, guidelines, recommendations, diagnosis, treatment, follow-up

## Abstract

Pancreatic ductal adenocarcinoma (PDAC) is the fourth leading cause of cancer-related death in women (7%) and the sixth in men (5%) in Italy, with a life expectancy of around 5% at 5 years. From 2010, the Italian Association of Medical Oncology (AIOM) developed national guidelines for several cancers. In this report, we report a summary of clinical recommendations of diagnosis, treatment and follow-up of PDAC, which may guide physicians in their current practice. A panel of AIOM experts in upper gastrointestinal cancer malignancies discussed the available scientific evidence supporting the clinical recommendations.

## 1. Introduction

According to AIRTUM (*Associazione Italiana dei Registri TUMori*), in 2019, about 13,500 new diagnoses (6800 male and 6700 female) of pancreatic ductal adenocarcinoma (PDAC) were expected in Italy, about 3% of all new diagnoses of cancer. The trend may be regarded as steady, given the 13,700 new cases in 2017. Data from the Italian Cancer Registries also show an increasing trend in the incidence of this disease among men [1]. PDAC is the fourth leading cause of cancer-related deaths in women (7%) and the sixth in men (5%) in Italy [1].

Cigarette smokers develop this disease 2 to 3 times more often than non-smokers do, with a reduction in the risk when people stop smoking [2].

Lifestyle and dietary factors also seem to be related to the risk of PDAC [3]. Chronic pancreatitis and diabetes mellitus are associated with a 10-fold and a 1.5–2-fold increase, respectively, in PDAC risk compared with the general population. Previous gastrectomy is also associated with a 3–5-fold increase in risk [4]. As regards genetic factors, approximately 3% to 10% of PDAC patients present a familial history [5]. This tumor is associated with several genetic syndromes, including hereditary pancreatitis syndrome, hereditary non-polyposis colorectal cancer, hereditary atypical multiple mole melanoma syndrome, and the Peutz–Jeghers syndrome [6]. Moreover, BRCA1/2 mutations are the most widespread causes of familial PDAC. BRCA2 and BRCA1 mutations lead to a risk of developing PDAC by a mean 2.26-fold and 3.5-foldincrease, respectively [5].

PDAC still maintains one of the worst prognoses among solid tumors, with a 5-year overall survival (OS) of 8% and a 10-year OS of 3%, achieving modest results with multimodal treatments in resectable settings [7,8]. Even though there is a trend for increased incidence, several trials in resectable/resected patients are showing improved survival rates. The multidisciplinary personalized, patient-centered therapies are moreover enhancing the combined treatments. So far, the Italian Medical Oncology Association (AIOM) has developed evidence-based guidelines in PDAC addressed to all specialists, and especially to oncologists involved in the management of these patients.

The aim of these guidelines is to standardize the multidisciplinary approach to PDAC by applying them to diagnostic and therapeutic care pathways in the regional cancer networks. All recommendations have been worked out on the basis of both up-to-date evidence from the literature and from the indications of the Italian Drug Agency (AIFA), which regulates the prescription of (antineoplastic) drugs.

## 2. Methods

The AIOM PDAC guidelineworking group includes the following physicians: oncologists, pancreatic surgeons, radiologists, gastroenterologists, pathologists and methodologists, working together with oncologist nurses and cancer affected patients. AIOM manual guidelines [9] report detailed methods for drafting such guidelines. Conferences or calls among authors were scheduled every 2 months to discuss clinical recommendations and review the best literature to disseminate. Every year, an updated version of the AIOM guidelines is published online on the AIOM website [10]. Recommendations addressed the most relevant clinical questions investigated according to PICO (Population, Intervention, Control, Outcome) methodology. The PICO question is considered according to specific clinical features (specific characteristics of disease, stage, etc.), treatment (the therapeutic intervention in question), potential alternatives to the treatment described (describing treatments considered as alternatives to the one in question), and considering the effect of measures and of primary and secondary outcomes by summarizing the evidence, making clinical recommendations, and degrees of strength of the recommendation in tabular form. A comprehensive, exhaustive, sensitive, and reproducible bibliographic search of the sources was previously carried out on various medical-scientific databases (PubMed, Embase, CENTRAL and area-specific databases). In the PubMed database, keywords were searched first through the MESH dictionary and then in "free search", using the diverse tools made available by the database with the use of search filters (age groups, type of study design, type of patients included and so on).

Based on the type of studies addressing the questions and their methods, AIOM guideline methodologists used the GRADE method (Grading of Recommendations Assessment, Development, and Evaluation) to classify the quality of each kind of evidence. In particular, the GRADE method assesses methodological bias within the studies; uniformity between different studies results; consistency of results across different studies; repeatability of results in a wider patient sample; the effectiveness of treatments. Treatment comparisons result in one out of four GRADE scores, reflecting the quality of the evidence: high-quality, moderate-quality, low-quality or very low-quality evidence (Table 1a). The strength of the recommendation is graded, based on clinical importance, on 4 levels (Table 1b). All questions (Q) are numbered and the clinical recommendations are summarized in tabular form (Table 2, Table 3, Table 4, Table 5 and Table 6).

### 2.1. Is Biopsy Always Indicated for Final Diagnosis in Patients with Clinical and Radiological Suspicion of PDAC, or Should It Be Considered Only in the Absence of Clear Radiological Signs of Malignancy and in Inoperable Patients?

In 2014, after a systematic review of several prospective and retrospective analyses, the consensus paper of the International Study Group for Pancreatic Surgery concluded that in patients with radiologic imaging strongly suspected for malignancy, who underwent surgery without preoperative histological diagnosis, the finding of benign neoplasia on the surgical specimen was between 5% and 13% [11]. A histological diagnosis of cancer was moreover reported in 5–9% of patients resected for chronic pancreatitis. In the preoperative diagnostic work-up, the endoscopic ultrasound approach to obtain a cytological or histological diagnosis is preferred to other methods [12].

In 2015, the European Federation of Society for Ultrasound in Medicine and Biology (EFSUMB) published guidelines for ultrasound procedures (INVUS), confirming the need for histopathological confirmation for advanced pancreatic cancer that is unsuitable for surgery and upfront to neoadjuvant treatments. Cytological or histological (biopsy) sampling can be an alternative according to indication and local protocols, and a combination of both is frequently used to improve diagnostic accuracy [74].

Although the specificity of the cytological diagnosis is very high, sensitivity is often low, and a diagnosis of cancer can only be excluded by surgery in a small number of patients.

The quality of the evidence extracted in a consensus statement [11], which supported a recommendation reliant on mainly retrospective moderate to low quality studies without control groups and an existing guideline [74,75], saw the application of a clear search strategy, selecting only recommendations with strong and widespread consensus among the drafters. Nevertheless, the panel of experts opted for a strong positive recommendation in view of the fact that any delays in diagnosis and surgery in patients affected by PDAC could reduce the chances of surgical success (Table 2).

In recent years, several studies have investigated the potential role of liquid biopsy in pancreatic cancer for early diagnosis. Circulating Tumor cells (CTCs), cell free DNA (cfDNA) and circulating microRNA (miRNA) can be detected in patients affected by pancreatic cancer with a rate ranging from 21% to 100% for CTCs, and with high specificity and sensitivity for miRNA panels (up to 100% and 90%, respectively) for PDAC diagnosis or high grade pancreatic intraepithelial lesions. Although the utility of liquid biopsy has been put forward, some concerns still exist regarding its extensive application in clinical practice.In the first place, there is lower sensitivity and specificity compared with traditional biopsies and secondly, there is a lack of consensus on the methodology for detection and assessment of CTCs, circulating cfDNA and miRNAs. Finally, availability is limited only at selected laboratories and there are relatively high costs associated with such advanced technology [76].

In conclusion, in patients with clinical and radiological suspicion of PDAC, pathological confirmation of diagnosis should be considered in the absence of clear signs of malignancy and in patients who are not candidates for surgery.

### 2.2. Can the Execution of a Contrast-Enhanced Multislice Chest and Abdomen Computed Tomography (CT) Be Considered the First Choice for Differential Diagnosis and Staging in Patients with a Pancreatic Mass Suspected for Adenocarcinoma?

CT diagnoses of PDAC have sensitivity and specificity from 70% to 100% [13]. So far it has always therefore been indicated for suspected PDAC. However, 27% of pancreatic adenocarcinomas of less than 2 cm are isodense in CT and hence, not directly identifiable [14]. The detection of pancreatic adenocarcinomas is reported to be superior for Magnetic Resonance (MR) imaging with respect to CT [15]. Choi et al. reported that MR was able to identify small PDAC better than CT [18] (Table 2).

So far, in patients with a pancreatic mass ≥ 2 cm suspected for adenocarcinoma, the execution of a contrast-enhanced multislice CT should be considered the first choice for differential diagnosis and staging.

### 2.3. Can MR Improve Liver Staging in Patients with Potentially Resectable PDAC?

Granata et al. compared MR, CT, and ultrasound with and without contrast on 35 patients with hepatic metastases. MR had the best diagnostic performance [16]. The limitations of this study are the retrospective methodology, the limited number of patients, and their heterogeneity. In patients with potentially resectable PDAC, MR with diffusion weighted sequences (DWI) significantly improved the diagnostic performance in the characterization of focal liver lesions, especially if small (≤1 cm), as compared with CT. In particular, MR after CT plays a role on liver staging [17] (Table 2).

For these reasons, MR could improve liver staging in patients with potentially resectable pancreatic adenocarcinoma.

### 2.4. Is Multislice CT Better than MR for Correct Definition of Non-Resectability in Patients Suspected for Locally Advanced PDAC?

CT has a positive predictive value of non-resectability ranging from 89% to 100% while the positive predictive value of resectability is lower (45–79%) [19]. A meta-analysis compared fiveimaging methods (CT, MR, angiography, PET and ultrasound) in the evaluation of suspected pancreatic masses [20]. Authors observed that CT and MR have similar sensitivity and specificity for defining vascular invasion. As regards evaluation of response to neoadjuvant treatments, the role of imaging is reduced [16,18] (Table 2).

So far, multislice CT is better than MR for correct definition of non-resectability in patients with pancreatic mass, suspected for locally advanced PDAC.

## 3. Treatment of Localized Disease

### 3.1. Is Preoperative Palliation of Jaundice Indicated in Patients with PDAC of the Head Scheduled for Pancreaticoduodenectomy?

In 2013, Fang et al. published a systematic revision and meta-analysis to assess the utility of preoperative biliary stent in jaundiced patients with resectableperiampullary tumors scheduled for pancreaticoduodenectomy [21]. Authors included six prospective randomized trials with 520 patients randomly assigned to the upfront surgery arm without biliary stent (255 patients) or to jaundice palliation (265 patients). Three primary outcomes were identified: postoperative mortality, major complications, and quality of life (QoL). Biliary drainage was either percutaneous or endoscopic. Bilirubin levels were between 40 and 250 micromol/dL (2.3–14.6 mg/dL). The rate of PDAC patients ranged between 60% and 95%. This meta-analysis showed no differences between the two groups regarding postoperative mortality (risk ratio 1.12, 95% CI 0.73 to 1.71; *p* = 0.60). Major postoperative complications (grade III–IV, according to Clavien-Dindo) were significantly more frequent in patients who underwent preoperative biliary drainage (599 complications per 1000 patients) compared with patients undergoing upfront surgery (361 complications per 1000 patients) (rate ratio 1.66, 95% CI 1.28 to 2.16; *p* < 0.001). Postoperative length of hospital stay did not differ significantly between the two groups (95% CI−1.28 to 11.02; *p* = 0.12).

This meta-analysis had several limitations, including the absence of data regarding QoL, the different tumor histotypes included in the studies, the lack of data regarding long-term survival, and the absence of comparison between endoscopic versus percutaneous stent (Table 3).

In conclusion, the preoperative palliation of jaundice should be avoided as it is associated with an increased risk of postoperative complications in patients with pancreatic head tumor and jaundice. Preoperative jaundice palliation should be limited to patients with cholangitis and/or high bilirubin levels (>15 mg/dL).

### 3.2. Is Extended Lymphadenectomy Associated with Improved Survival Compared to Standard Lymphadenectomy in Patients with PDAC Undergoing Surgical Resection?

Dasari et al. published the results of a meta-analysis that analyzed five randomized clinical trials comparing extended versus standard lymphadenectomy in PDAC patients undergoing pancreaticoduodenectomy [22]. The primary endpoint was survival. More than 500 patients were analyzed including those who received extended (N = 276, 50.1%) or standard lymphadenectomy (N = 270, 49.9%). The mean number of resected nodes ranged between 13.3 and 17.3 for standard lymphadenectomy and between 19.8 and 40.1 for extended lymphadenectomy. Extended lymphadenectomy was associated with a significantly higher number of excised lymph nodes compared with standard lymphadenectomy (mean difference=15.73, 95% CI = 9.41–22.04; *p* < 0.00001; heterogeneity among different studies was high, I^2^ 88%). Lymph node metastases were reported in 58–68% and in 55–70% of patients who underwent extended and standard lymphadenectomy, respectively. This meta-analysis did not identify a higher rate of lymph node metastases in patients undergoing extended lymphadenectomy (OR = 0.78, 95% CI = 0.55–1.10; *p* = 0.16). Moreover, extended lymphadenectomy did not significantly improve survival compared with standard procedures (HR = 0.88, 95% CI = 0.75–1.03; *p* = 0.11). On the contrary, extended lymphadenectomy was associated with a higher risk of overall postoperative morbidity (relative risk = 1.23; 95% CI = 1.01–1.50; *p* = 0.004; I^2^: 9%). In conclusion, as compared with standard lymphadenectomy, extended lymphadenectomy is associated with an increased risk of postoperative complications without improving patient survival rates. 

Even if the meta-analysis presented intriguing results, there is high heterogeneity, including the number of excised lymph nodes and the extension of lymph node clearance, in patients undergoing extended lymphadenectomy in studies from Japan/Korea compared with studies from US/Europe. Moreover, the role of extended/standard lymphadenectomy has been evaluated only for PDAC of the head but not for those of the body-tail (Table 3).

### 3.3. Should Pancreatic Surgery Be Performed in High-Volume Centers in Order to Decrease Postoperative Morbidity and Mortality in Resectable PDAC Patients?

Hata et al. published a meta-analysis evaluating the influence of volume of pancreatic surgery performed in high-volume versus low-volume centers on postoperative morbidity and mortality following pancreaticoduodenectomy [23]. Authors identified 13 studies that analyzed national databases from 11 different countries with 58,023 patients between 1990 and 2010. The cutoff value used to identify a high-volume center ranged between 10 and 54 pancreaticoduodenectomies/year, with a median value of 20. The rate of high-volume centers ranged between 3% and 64% in different studies. The overall pooled odds ratio (OR) for postoperative mortality was 2.37, favoring high-volume centers (95% CI: 1.95–2.88) with high heterogeneity (I^2^: 63%). Authors performed a sensitivity analysis without the three studies with the higher level of heterogeneity, and this sub-analysis showed a pooled OR of 2.04 (95% CI: 1.79–2.33), in favor of high-volume centers (I^2^: 0%; *p* < 0.0001). Authors defined different categories based on the number of procedures performed every year. Pooled OR for postoperative mortality was 1.94 for the category 1–19 PD/year, 2.34 for the category 20–29 PD/year and of 4.05 for the subgroup with more than 30 PD/year. The mean postoperative length of hospital stay was 3.3 days shorter for patients who underwent surgery at high-volume centers (95% CI: 1.98–4.55, *p* < 0.00001) with moderate heterogeneity for this analysis (I^2^: 43%).Study limitations included significant differences among the studies for the definition of high-volume centers and for the characteristics of patients who underwent surgery (i.e., age and associated comorbidities), differences among the healthcare systems of different countries involved in the studies (i.e., Asia, Europe, USA), and the high or moderate heterogeneity for different outcomes analysis.

Balzano et al. published the results of pancreatic surgery procedures for PDAC performed in Italy between 2010 and 2012 [24]. Hospitals were divided into fivegroups based on the number of resections performed using quintiles as the cutoffs. In the study period, 10,936 procedures were performed in 544 hospitals. Hospitals were divided as follows: very low volume (mean 1.5 procedures, three-year range: 0–9; 408 hospitals, 75%), low volume (mean 5.5 procedures, three-year range: 10–26; 76 hospitals, 14%), intermediate volume (mean 13.5 procedures, three-year range: 27–57; 37 hospitals, 6.8%), high-volume (mean 33.5 procedures, three-year range: 58–141; 17 hospitals, 3.1%), very high volume (mean 91 procedures, minimum number of procedures performed in threeyears > 141; 6 hospitals, 1.1%). The probability of undergoing an explorative/palliative operation was inversely correlated with the volume (24% in very high-volume centers compared with 62.5% in very low-volume centers; OR—5.175), while the rate of pancreatic resections increased from 46.1% in very low-volume centers to 86.9% in very high-volume centers (OR—7.429). Median postoperative mortality following pancreatectomy was 11.7% in very low-volume centers, 8.9% in low-volume centers, 6.6% in intermediate volume centers, and then,5% and 3.8% in high-volume and very high-volume centers, respectively. Although the meta-analysis showed that surgery-related mortality significantly differs based on different centers experience, there are several limitations including “significant differences across the studies for the definition of high-volume center and for the characteristics of patients who underwent surgery (i.e., age and associated comorbidities), differences between healthcare systems of different countries involved in the studies (i.e., Asia, Europe, USA), and the high or moderate heterogeneity for different outcomes analysis”. For these reasons, the overall quality of evidence was low. In any case, pancreatic resection for PDAC should be performed in high/very high-volume centers in order to decrease perioperative morbidity (Table 3).

### 3.4. Does Postoperative Chemoradiotherapy (Administered with a Total Dose of Radiotherapy of at Least 50 Gy in Combination with Gemcitabine or Fluoropyrimidine) Obtain a Benefit in Terms of Survival in Patients with Resected Stage Ia-III PDAC (R0-R1) with a Karnofsky Performance Status (PS) of at Least 50%?

The randomized ESPAC-1 study assessed (with 2 × 2 factorial design) the efficacy of postoperative systemic chemotherapy versus no chemotherapy and the efficacy of chemoradiotherapy with 5FU as a radiosensitizer versus no chemoradiotherapy in 289 PDAC patients who underwent R0-R1 surgery [25]. The primary endpoint of the study was OS. There was a statistically significant advantage in terms of OS in patients who received systemic chemotherapy with or without chemoradiotherapy compared to those who did not receive systemic chemotherapy (median OS, 20.1 months vs. 15.5 months, respectively, *p* = 0.009). In contrast, the survival of patients who received chemoradiotherapy alone or in combination with systemic chemotherapy was worse compared with patients who received systemic chemotherapy or observation (median OS, 15.9 months vs. 17.9 months, respectively, *p* = 0.05). Although the study was not sized to compare the various subgroups, no statistically significant differences were observed among the four treatment arms. According to these results, postoperative chemoradiotherapy was associated with a detrimental effect on survival. The study has several limitations, including the different timing of the treatments (systemic chemotherapy was delayed in the arm treated with chemoradiotherapy followed by chemotherapy compared to the group treated with chemotherapy alone) and the modality of chemoradiotherapy treatment. Analyzing the aspects related to the radiotherapy treatment, it should be noted that the absence of quality control of the treatment, the ’split course’ scheme, the inappropriate total dose of 40 Gy administered, and the use of Cobalt 60 and obsolete 2D radiotherapy techniques means the results of this study are not applicable in clinical practice. In addition, the activity of 5FUadministered in bolus as a radiosensitizer is lower than that of 5FU given in continuous infusion and compared to gemcitabine.

The meta-analysis published by Stocken et al. collected data from fiverandomized clinical trials to investigate the role of chemotherapy and chemoradiotherapy in the adjuvant setting on 875 PDAC patients [26]. The primary endpoint was OS. In particular, the authors observed that adjuvant chemotherapy induced a reduction in the risk of death by 25% (HR 0.75; 95% CI: 0.64, 0.90, *p* = 0.001) with a median OS of 19 months (95% CI: 16.4, 21.1) compared to 13.5 months (95% CI: 12.2, 15.8) of untreated patients. OS at two and five years was 38% and 19%, respectively, compared to 28% and 12% in the untreated patients group. The meta-analysis does not show a significant reduction in the risk of death in patients treated with chemoradiotherapy (HR 1.09; 95% CI: 0.89, 1.32, *p* = 0.43), with a median OS of 15.8 months (95% CI: 13.9, 18.1) compared with 15.2 months (95% CI: 13.1, 18.2) in untreated patients. A subgroup analysis showed that chemoradiotherapy is more effective than chemotherapy in patients operated with R1 surgery (χ^2^ = 4.2, *p* = 0.04). An important limitation of the meta-analysis is that the studies with radiotherapy all used a ’split course’ radiotherapy scheme, an inadequate total dose of 40 Gy, Cobalt60, and obsolete radiotherapy techniques.

A randomized phase II study assessed four cycles of gemcitabine (control arm) or gemcitabine for two cycles followed by weekly gemcitabine in combination with radiotherapy (total dose 50.4 Gy) after surgical resection [27]. Ninety patients were enrolled (45 per arm): 86.7% and 73.3% completed the adjuvant treatment, respectively (95% CI: 58.1–85.4%). Only three cases of grade 3 or higher toxicity were observed in the experimental arm. Median disease-free survival (DFS), the secondary endpoint of the study, was 12 months and 11 months in the experimental arms and chemotherapy alone, respectively. Local recurrence rates were 11% and 24% in the combination and control arm, respectively. It should however be noted that all patients included in the group received an R0 resection, therefore, the effect on local disease control could be underestimated as compared with other adjuvant therapy studies.

The phase III study RTOG 9704 investigated the role of adjuvant gemcitabine (1000 mg/m^2^/weekly; N = 230) or 5FU (as continuous infusion 250 mg/m^2^/day; N = 221) 3 weeks before and 12 weeks after radiotherapy (total dose 50.4 Gy;1.8 Gy/day) simultaneously with 5FU (250 mg/m^2^/day continuous infusion) in 451 PDAC patients [28]. The primary endpoint was OS. A median survival of 20.5 months and a three-year survival of 31% in the gemcitabine group vs. a median survival of 16.9 months and a three-year survival of 22% in the 5FU group (hazard ratio, 0.82 [95% confidence interval, 0.65–1.03]; *p* = 0.09) were observed. An update of the five-year RTOG 9704 data confirmed the absence of statistically significant differences between the two groups, although patients with PDAC of the head continued to show a positive trend in terms of absolute survival in the gemcitabine arm (*p* = 0.08) [29] (Table 3).

### 3.5. Could an Adjuvant Regimen with mFOLFIRINOX Be Considered Superior to Gemcitabine Alone in Terms of DFS in Patients with Radically Resected (R0 or R1) PDAC with ECOG PS 0 or 1 and Age ≥ 18 and <70 Years?

The French randomized phase III PRODIGE24/ACCORD6 study randomized 493 patients with radically resected PDAC (R0/R1), ECOG PS between 0 and 1 and aged between 18 and 70 years to receive an adjuvant treatment with modified FOLFIRINOX vs. gemcitabine monotherapy [30]. Gemcitabine was administered according to the classic schedule (1000 mg/m^2^ on days 1,8, and 15 every 28 days), while the mFOLFIRINOX arm provided a dose reduction in irinotecan (performed after a safety analysis) and omission of 5FU bolus compared to the original schedule used in metastatic setting (oxaliplatin 85 mg/m^2^, leucovorin 400 mg/m^2^, irinotecan150 mg/m^2^ and 5FU 2400 mg/m^2^c.i. for 46 h, every 14 days). The expected duration of treatment was six months in both arms. Four hundred and ninety-three patients (247 in the mFOLFIRINOX arm and 246 in the gemcitabine arm) were enrolled and stratified by center, lymph node involvement, resection margin status, and postoperative levels of CA19-9 (≤90 U/mL vs. 91–180 U/mL). The primary endpoint was mDFS, while the secondary endpoints were OS, DFS and toxicity profile. Demographic and disease characteristics were balanced between the two treatment arms, with the exception of lymphovascularinvasion, more frequent in the mFOLFIRINOX arm (73.7% vs. 63.1%, *p* = 0.02). At a median follow-up of 33.6 months, median DFS was 21.6 months in the mFOLFIRINOX arm and 12.8 months in the gemcitabine arm (HR 0.58; 95% CI: 0.46–0.73; *p* < 0.001). The median OS was 54.4 months in the mFOLFIRINOX arm vs. 35 months for gemcitabine (HR 0.64, CI 95%: 0.48–0.86, *p* = 0.003), while median DFS was 30.4 months vs. 17.7 months (HR 0.59). In subgroup analyses, the advantage of mFOLFIRINOX was maintained in all subgroups. In particular, the benefit of the experimental regimen was similar among patients <65 years of age and patients ≥65 years of age. However, in 101 patients ≥70 years of age (20.5%), the DFS advantage of mFOLFIRINOX over gemcitabine did not reach a statistical significance (HR 0.86; 95% CI: 0.53–1.39). The incidence of grade 3–4 adverse events was higher for the experimental arm versus gemcitabine alone (75.9% vs. 52.9%), especially for diarrhea (18.9% vs. 3.7%), fatigue (11% vs. 4.6%), mucositis (2.5% vs. 0%), peripheral neuropathy (9.3% vs. 0%), and vomiting (5.1% vs. 1.2%). Only 66.4% of the patients in the mFOLFIRINOX arm completed the six months of therapy, as opposed to 79% of those enrolled in the gemcitabine arm (*p* = 0.002). This interruption was attributable to toxicity in 8.8% of cases in the experimental arm. One toxic death was recorded in the control arm (interstitial pneumonia).

The strength of this study is the correct selection of patients, with a post-surgical mandatory CT scan and the exclusion of cases with high post-resection levels of CA19-9 (>180 U/mL). This trial demonstrates a large advantage for mFOLFIRINOX as compared with gemcitabine monotherapy, but it is important to consider the high proportion of non-hematological adverse events in the experimental arm, and the fact that slow accrual (approximately 1.5 patients per center per year) suggests careful patient selection. These considerations underline the fact that caution should be exercised in transferring the results to the whole population of radically resected patients. It should also be considered that follow-up is excessively short for this type of patient, with a very low number of patients exposed to risk after 30 months, making medium-long term survival results unreliable. Furthermore, the median survival was incorrectly calculated by estimating it from the Kaplan–Meier actuarial curves, and so, the correct value is not known. In conclusion, these data support a modification of clinical practice, placing mFOLFIRINOX as the adjuvant treatment of choice in carefully selected patients, <70 years of age and with good ECOG PS (Table 3).

### 3.6. Could an Adjuvant Regimen with nab-Paclitaxel Plus Gemcitabine be Considered Superior to Gemcitabine Alone in Terms of DFS in Patients with Macroscopic Completed Resected PDAC with ECOG PS 0 or 1 and Age ≥18?

The phase III APACT study, presented at ASCO 2019 by Tempero et al., enrolled 866 patients (median age 64 years) with histologically confirmed PDAC, macroscopic complete resection, ECOG PS 0 or 1, and CA19-9 < 100 U/mL to receive within 12 weeks from surgery, nab-paclitaxel (125 mg/m^2^) plus gemcitabine (1000 mg/m^2^)on days 1, 8, and 15 of six 28-day cycles vs. gemcitabine alone (1000 mg/m^2^ on days 1, 8, and 15 of six 28-day cycles), for six cycles in each arm [77]. The primary endpoint was independently assessed radiologic DFS. The secondary endpoints were OS and safety. Stratification factors were resection status (R0/R1), lymph node status (N0/N+), and geographic region. Most patients had lymph-node positive disease (72%) and received R0 resections (76%). With a median follow-up of 38.5 months, median DFS by independent review was 19.4 months with combination vs. 18.8 months with gemcitabine alone (HR = 0.88; *p* = 0.1824). The benefit was greater in patients with G2 tumors, N-positive disease, R1 resection, and normal CA19-9 level.

For the prespecified analysis of investigator-assessed DFS, a benefit was shown for the combination, with a median DFS of 16.6 months for doublet vs. 13.7 months for single agent (HR = 0.82; *p* = 0.0168). Data on OS are still immature, but at data cutoff, interim median OS was 40.5 months for gemcitabine plus nab-paclitaxel vs. 36.2 months for gemcitabine (HR = 0.82; *p* = 0.045). Regarding toxicities, grade ≥ 3 adverse events were reported in 86% vs. 68% of patients with combination vs. gemcitabine;most common grade ≥ 3 adverse events were neutropenia (49% vs. 43%, respectively) and fatigue (10% vs. 3%).

The study failed to demonstrate a benefit for combination over gemcitabine in adjuvant setting, as regards the primary endpoint of independently-assessed DFS. It is important to note that DFS per investigator assessment showed a prolonged DFS for nab-paclitaxel plus gemcitabine, and these results could be a caveat for future clinical trial design. This trial is in fact the first adjuvant study in which independently-assessed radiologic DFS was used as the primary endpoint. Discordances between central and local assessment could be explained by the different way of determining progressive disease by central review, which was based on CT scan only, and by local investigators, which also included clinical deterioration, pain, rise in CA19-9 levels, tumor biopsy, MR, or FDG-PET scan. Overall, independently-assessed radiologic DFS does not reflect what happens in clinical practice and has serious limitations for being regarded as a reliable surrogate of OS and, accordingly, an adequate primary endpoint. In addition, OS data are still immature and the study is not yet fully published. It is therefore not yet possible to draw conclusions or formulate an evidence-based clinical recommendation. To date, due to the formally negative results of the primary endpoint of the trial and the absence of fully published data, the use of gemcitabine plus nab-paclitaxel in the adjuvant setting is not recommended.

### 3.7. Could Neoadjuvant Treatment (Chemotherapy or Chemoradiotherapy) Improve Oncological and Surgical Outcomes in Patients with Resectable PDAC without Contraindications to Surgery?

A recent meta-analysis considered 14 prospective clinical studies (including phase 1, phase 2, and phase 3) on 616 patients with resectable PDAC who received neoadjuvant therapy [31]. Among these patients, regardless of the type of therapy administered, 1.8% (95% CI: 1.0–3.4%) had a complete response [I2 = 0.0% (*n* = 13)], 14.6% (95% CI: 7.5–26.4%) had a partial response [I2 = 44.3% (*n* = 12)], 62.2% (95% CI: 46.5–75.7%) had a stable disease [I2 = 45.7% (*n* = 12)], and 13.4% (95% CI: 8.4–20.7%) had disease progression [I2 = 36.0% (*n* = 12)]. Furthermore, 73.0% (95% CI: 64.8–79.9%) [I2 = 40.7% (*n* = 14)] of these patients had a resection and 88.2% (95% CI: 82.1–92.5%) of the resected cases had microscopic disease-free margins [I2 = 34.2% (*n* = 12)]. OS was 17.7 months (95% CI: 9.4–27.2 months), ranging from 25.2 months (95% CI: 11.7–34 months) for patients who were resected to 8.8 months (95% CI: 7.1–11 months) for those who did not undergo resection.

Considering the limits of the studies included in this metanalysis, the clinical value of this approach is still unclear. In particular, the sample size was small and there were no control groups, the definitions of resectability were heterogeneous and the treatment procedures were different. Indeed, many studies were prematurely interrupted due to slow recruitment and preliminary results were underpowered and difficult to interpret. Furthermore, the greatest criticism that can be made of these studies was the use of outdated chemotherapy regimens. Moreover, the primary endpoint of some studies was to improve local control of the disease with chemoradiotherapy. Finally, the meta-analysis included six studies, in which patients with locally advanced and borderline resectable disease were also eligible.

More recently, Reni et al. reported the results of a randomized phase2 study conducted in nine Italian cancer centers [32]. Eighty-eight patients with histologically proven resectable PDAC were randomized into three groups: 26 patients underwent surgery followed by adjuvant chemotherapy with gemcitabine (1000 mg/m^2^ on days 1, 8, and 15 every four weeks) for six cycles (group A); 30 received surgery followed by six cycles of adjuvant chemotherapy according to the PEXG regimen (cisplatin 30 mg/m^2^, epirubicin 30 mg/m^2^, and gemcitabine 800 mg/m^2^ on days 1 and 15, every four weeks and capecitabine 1250 mg/m^2^ on days 1–28, group B) (group B); 32 received three cycles of PEXG before and three cycles after surgery (group C). The primary endpoint was the percentage of patients without events (progression, relapse, new tumor appearance, distant metastasis or death) at oneyear from randomization. Six patients in group A (23%, 95%CI 7–39) were event-free at one year, as were 15 in group B (50%, 95% CI 32–68%) and 19 (66%, 95% CI 49–83%) in group C. The median OS were 20.4 months (95% CI 14.6 to 25.8) (group A), 26.4 months (95% CI 15.8–26.7) (group B), and 38.2 months (95% IC 27.3–49.1) (group C). The main grade 3 toxicities consisted of neutropenia (28% in group A, 38% in group B, and in group C 28% and 48% before and after surgery, respectively) and anemia (6% in group A, 19% in group B, and for group C, 28% and 24% before and after surgery, respectively). No treatment-related deaths were observed. This randomized phase 2 study provides evidence of the efficacy of neoadjuvant chemotherapy in resectable PDAC. Although a confirmatory phase 3 study is needed, perioperative treatment with PEXG may improve surgical and oncological outcomes. Moreover, other chemotherapy combinations could also enhance survival in these sets of patients (Table 3).

## 4. Treatment of Borderline Resectable Pancreatic Cancer and Locally Advanced Pancreatic Cancer

### 4.1. Does Preoperative Treatment Increase Survival in Comparison with Immediate Surgery in Borderline Resectable Pancreatic Cancer (BRPC)?

Though diverse definitions of BRPC have been put forward, it generally refers to a PDAC that is not metastatic but that is associated with a high risk of incomplete surgical resection or of early recurrence in case of immediate surgery [33]. In light of the difficulties in defining this clinical condition, patients with BRPC are often included in clinical trials together either with patients with resectable or with those with locally advanced disease [8]. Several meta-analyses have been carried out on this subject but none of them have included any randomized trial (Table 7).

A comprehensive meta-analysis published by the Dutch Pancreatic Cancer Group evaluated the role of preoperative treatment in resectable and BRPC [34]. The authors included three randomized trials, 12 prospective and 14 retrospective cohort studies. The three randomized trials considered only patients with resectable PDAC and only two of the three randomized trials enrolled patients that were candidates for upfront surgery or preoperative therapy. Neoadjuvant treatments were heterogeneous across trials. In the majority of trials, radiotherapy was included in preoperative treatment and adjuvant chemotherapy was suggested after pancreatic resection for patients in both arms. Regarding the topic of BRPC therapeutic management, the meta-analysis assessed 927 patients treated with immediate surgery versus 881 patients treated preoperatively. Surgical resection was performed in 85.3% (82.9–87.5%) of patients assigned to upfront surgery and in 65% (61.8–68.2%) of those treated with neoadjuvant therapy. Radical resection was obtained more frequently after neoadjuvant therapy (88.6%) than with surgery alone (63.9%). Overall, the percentage of patients who underwent R0 radical resection was similar in both groups (55% vs. 58%). Toxicity was related to the type of neoadjuvant therapy (chemotherapy or chemoradiotherapy) and mainly consisted of gastrointestinal or hematological effects with superimposable rates when compared to adjuvant setting. Preoperative therapy did not give increased surgical morbidity or mortality. Survival was longer in patients treated with a neoadjuvant approach, with a median OS of 19.2 months (range 11–32) compared to 12.8 months (11.6–16.3) obtained with upfront surgery. According to the modality of neoadjuvant therapy received, mOS was 20.9 months (13.6–27.2 months) with chemotherapy alone and 17.8 months (9.4–32 months) with chemoradiotherapy.

Similar results in terms of survival and resection rate have been reported in a more recent meta-analysis indicating that adding radiotherapy to neoadjuvant chemotherapy had little effect on OS [35]. 

Another meta-analysis included 24 studies (8 prospective, 16 retrospective), considering only BRPC treated with neoadjuvant FOLFIRINOX [36]. The study assessed 313 patients with patient-level survival data obtained for 283 patients in 20 studies and showed favorable results in terms of survival and resection rates [36]. An analysis from National Cancer Database including 1980 patients with BRPC showed improved mOS (25.7 months vs. 19.6 months, *p* < 0.0001) and signs of pathological response (reduction in node-positivity and in margin-positive resection) with neoadjuvant therapy versus surgery followed by adjuvant treatment [37].

Recently, two multicenter randomized studies have been published [38,39]. The Korean phase II–III study considered BRPC patients randomized to receive chemoradiotherapy with gemcitabine and a dose of 54 Gy of radiotherapy before or after surgery, followed by adjuvant chemotherapy with gemcitabine [38]. This trial was prematurely stopped after interim analysis on the first 57 enrolled patients (of the 110 expected) demonstrated the superiority of preoperative chemoradiotherapy. Neoadjuvantchemoradiotherapy resulted superior to postoperative treatment with a mOS of 21 months vs. 12 months and 40.7% of patients alive at two years versus 26.1%. Preoperative treatment increased R0 resections to 51.8% vs. 26.1% obtained with immediate surgery. Tumors resected after chemoradiotherapy were smaller in diameter and more frequently node negative. No differences in terms of toxicity and surgical morbidity or mortality were observed. In the Dutch PREOPANC trial, 246 patients were randomized to receive preoperative chemoradiotherapy with gemcitabine or immediate surgery; 113 patients had BRPC [39]. Considering only BRPC patients, preoperative chemoradiotherapy increased OS and DFS compared with immediate surgery (HR 0.62, 95%CI: 0.40–0.95, *p* = 0.029 for OS, HR 0.59, 95%CI: 0.39–0.89, *p* = 0.013 for DFS) and improved R0 resection rate (79% vs. 13%, *p* < 0.001) without seriously affecting safety (Table 4).

### 4.2. Is Initial Chemotherapy Versus Chemoradiotherapy Recommended in Patients with Locally Advanced Pancreatic Cancer (LAPC)?

A recent literature review and meta-analysis evaluated five randomized trials and three observational studies for a total of 832 patients, 593 in randomized trials and 239 in observational trials [40]. All studies included a comparison of sole chemotherapy treatment with initial chemoradiotherapy treatment, except for the LAP-07 study, in which an initial sole chemotherapeutic treatment was performed in both arms, followed or not followed by chemoradiotherapy. Chemotherapy regimens consisted of a gemcitabine-based regimen in six studies (three randomized and three observational) or 5FU-based regimen in the remaining two randomized trials. The dose of radiotherapy used was more than 50 Gy in six studies, 45 Gy in one study and 40 Gy in another. Radiotherapy was administered by 3D conformational techniques in six out of eight studies. The median age of patients in the different studies ranged from 60 to 68 years. ECOG PS was 0–1 in a percentage of patients between 80% and 100%. The use of initial chemoradiation did not produce a significant improvement in OS (HR 0.87; 95% CI, 0.63–1.21, *p* = 0.41) or in DFS (HR 0.90; 95% CI, 0.74–1.10; *p* = 0.30) in the five randomized trials analyzed. Conversely, in the three observational studies, an advantage in terms of OS (HR 0.48; 95% CI, 0.35–0.66, *p* < 0.001) and DFS (HR 0.58; 95% CI, 0.37–0.92; *p* = 0.02) for patients treated with chemoradiotherapy was reported. It should be considered that the quality of the studies on the various outcomes measured was very low. In both randomized trials (HR 3.3; 95% CI, 1.12–14.22; *p* = 0.03) and observational studies (HR 3.59, 95% CI, 0.18–71.37, *p* = 0.40), chemoradiotherapy was associated with an increased risk of adverse events, such as grade 3 or 4 diarrhea, nausea or vomiting (randomized studies: HR 2.53; 95% CI, 1.31–4.87; *p* = 0.006; observational studies: HR 3.59; 95% CI, 0.43–29.74; *p* = 0.24). These data do not seem to support addition of initial radiotherapy to chemotherapy in this subset of patients (Table 4).

So far, chemotherapy may be considered as a first-choice option as initial therapy, as an alternative to chemoradiotherapy in patients with LAPC.

### 4.3. Is Consolidation Chemoradiotherapy Indicated in Progression-Free Patients after Induction Chemotherapy in LAPC?

In 2009, Huguet et al. published a systematic qualitative review of comparative literature on 21 studies (2 meta-analyses, 13 randomized trials and 6 non-randomized trials) involving 1854 LAPC patients [41]. The role of radiotherapy was assessed in relation to the following therapeutic options: chemoradiotherapyvs. best supportive care; chemoradiotherapyvs. exclusive radiotherapy; chemoradiotherapyvs. chemotherapy; induction chemotherapy followed by chemoradiotherapyvs. chemoradiotherapyvs. chemotherapy. Concomitant chemoradiotherapy was superior to the best supportive therapy in terms of OS (13.2 months vs. 6.4 months, *p* < 0.001) and of QoL (*p* < 0.001). Concomitant chemoradiotherapy prolonged survival compared to radiotherapy alone (HR = 0.69; 95% Cl, 0.51–0.94). OS was not significantly different after chemoradiotherapy or chemotherapy alone in studies that evaluated the comparison between the two therapeutic approaches (HR = 0.79; 95% CI, 0.32–1.95). A further limitation of this study was the heterogeneity of the analyzed data.

LAP07 randomized trial evaluated 442 patients with LAPC who underwent a first randomization to gemcitabine chemotherapy alone or associated with erlotinib [42]. After four months, 269 patients (61%) with a controlled disease were further randomized to chemoradiotherapy for two months or to continuation of the same chemotherapy regimen. The median follow-up was 36.7 months (95% CI, 27.6–44.2 months). As regards the first randomization (chemotherapy with gemcitabine vs. gemcitabine-erlotinib), no differences in mOS and in mPFS were found between the two arms (HR, 1.12; 95% CI, 0.92–1.36; *p* = 0.26). Additionally, in the second randomization (chemoradiotherapyvs. chemotherapy), no differences were observed in terms of mOS (15.2 months vs. 16.5 months; HR 1.03; 95% CI, 0.79–1.34; *p* = 0.83) and mPFS (9.9 months vs. 8.4 months; HR, 0.78; 95% CI, 0.61–1.01; *p* = 0.06). Radiotherapy was administered up to the total dose of 54 Gy in combination with capecitabine. Gemcitabine in combination with erlotinib was associated with a shorter OS than gemcitabine alone (14.5 months vs. 17.1 months; HR, 1.32; 95% CI, 1.01–1.72; *p* = 0.04). Among patients who underwent the second randomization, local progression was less frequent in the chemoradiotherapy arm when compared with the chemotherapy arm (32% vs. 46%), while distance progression was higher (60% vs. 44%). The time interval without therapy was longer in patients receiving chemoradiotherapy (6.1 months vs. 3.7 months, *p* = 0.02). Treatment tolerance was equivalent in the two arms. The greatest toxicities were recorded in gemcitabine plus erlotinib arm, with the exception of nausea. The study therefore shows that chemoradiotherapy produced a nearly significant increase in PFS, a significant delay in the initiation of subsequent therapy, and significantly better local control, even if it did not lead to a significant survival benefit. Its main limitations are the double randomization and the use of non-optimal chemotherapy regimen.

A comprehensive retrospective analysis of the American National Cancer Database considered over 5000 patients treated with exclusive chemotherapy or followed by chemoradiotherapy, coupled through propensity-score analysis [43]. The use of chemoradiotherapy lead to superior mOS (12.3 months vs. 9.8 months) and two-year OS (16.3% vs. 12.9%). Regarding the schedule of induction treatment, a polychemotherapy treatment was superior in terms of survival compared with monotherapy (HR 0.71; 0.68–0.74; *p* < 0.001) (Table 4). Table 7 reports a summary of the studies.

## 5. Treatment of Advanced Disease

### 5.1. First Line

#### 5.1.1. Does a FirstLine Chemotherapy Treatment with 3 or 4 Drugs Increase Survival in Patients with Metastatic PDAC, Karnofsky PS > 70 or ECOG PS ≥ 1, and Age ≤ 70 Years?

A phase III clinical trial conducted by Reni et al. evaluated 99 patients with metastatic PDAC, KPS> 70, and age ≤ 70 years, a statistically significant advantage in terms of PFS and OS related to the use of a four-drug regimen (cisplatin, epirubicin, gemcitabine, and 5FU) was observed when compared to gemcitabine monotherapy [44]. Four-months and mPFS were 60% vs. 28% (HR 0.46 [0.26–0.79]; *p* = 0.001) and 5.4 months vs. 3.3 months (HR 0.51 [0.33–0.68]; *p* = 0.0033) in the combination and gemcitabine arm, respectively. The one-year survival rate was 38.5% vs. 21.3% (HR 0.65 [0.42–1.09], *p* = 0.047), respectively. The observed toxicities were higher in the combination arm. In particular, grade 3–4 neutropenia was 43% in the combination arm as compared with 16% in the monotherapy (*p* < 0.0001), and grade 3–4 thrombocytopenia was 30% in the combination arm compared with 1% in the monotherapy (*p* < 0.0001). There was no impact on patients’ QoL even though the sample size was underpowered to provide statistical evidence.

A French randomized phase III study analyzed 342 patients, between 18 and 75 years old, affected by metastatic PDAC with ECOG PS between 0 and 1, randomly assigned to receive first line FOLFIRINOX or gemcitabine in monotherapy [45]. FOLFIRINOX yielded a significant advantage compared to gemcitabine alone in terms of PFS (6.4 months vs. 3.3 months; HR 0.47 [0.37–0.59]; *p* < 0.0001) and OS (11.1 months vs. 6.8 months; HR 0.57 [0.45–0.73]; *p* = 0.001 — 1-yOS 48.4% vs. 20.6%). However, the extra-hematological toxicity profile grade 3–4 of FOLFIRINOX (asthenia 23%, vomit 15%, diarrhea 13%, peripheral neuropathy 9%) was not completely acceptable for a palliative treatment of this malignancy. Moreover, the remarkable treatment burden for the patient must also be taken into account (i.e., four hospital visits per month without considering adverse events, the need to implant a central venous catheter for the administration of 5FU, and the systematic use of growth factors). Of note, no significant differences between the two arms were observed in the Global Health Status and QoL scales, except for those who had grade 3–4 diarrhea during the first eightcycles of FOLFIRINOX. After six months, 31% of the patients in the FOLFIRINOX group had a definitive decrease in overall health and QoL scores compared to 66% in the gemcitabine group (hazard ratio, 0.47, 95% CI, 0.30 to 0.70; *p* < 0.001). Overall, the results of this study should be interpreted with caution due to the patient selection, suggested by the fact that this trial took four years for 48 centers to enroll 342 patients. Neither of the studies were blinded because of monotherapy control. Although these studies included patients of advanced age, the sample size of the population of patients over 65–70 years of age was limited, and well-selected due to the PS cutoff, and cannot be deemed representative of elderly patients in the general population (Table 5).

#### 5.1.2. Does a FirstLine Chemotherapy Treatment with Gemcitabine/nab-Paclitaxel Combination Increase Survival in Patients with Metastatic PDAC, Karnofsky PS ≥ 70 or ECOG PS ≥ 1, and Age> 18 Years?

A multicenter phase III study evaluated 861 patients with metastatic PDAC, aged over 18 years and Karnofsky PS ≥ 70, randomized to receive firstline treatment with gemcitabine alone (430 patients) vs. gemcitabine plus nab-paclitaxel (431 patients) [46]. The combination regimen was associated with an increase in both PFS (5.5 months vs. 3.7 months; HR 0.69 [0.62–0.83]; *p* < 0.0001) and OS (8.5 months vs. 6.7 months; HR 0.72 [0.62–0.83]; *p* < 0.0001) compared to gemcitabine alone. In this study, 4-year survival of 4% was also observed in the combination therapy arm compared with the absence of long-term survivors among patients treated with gemcitabine alone. The incidence of grade ≥ 3 toxicity was higher in the combination arm (38% vs. 27% neutropenia, 17% vs. 7% neutropenia, and 17% vs. 1% neuropathy). Neuropathy was rapidly reversible, allowing treatment to resume its course in 44% of cases. The survival benefit of this combination outweighs the risks of possible damage, given the reversibility of side effects especially of neuropathy. A limiting factor is that it was not a blind trial (Table 5).

#### 5.1.3. Does FirstLine Chemotherapy Treatment with PAXG Combination Increase Survival in Patients with Metastatic PDAC, Karnofsky PS ≥ 70 or ECOG PS ≥ 1, and Age 18–75 Years?

A further therapeutic option was evaluated with a four-drug regimen, PAXG (cisplatin 30 mg/m^2^, nab-paclitaxel 150 mg/m^2^ and gemcitabine 800 mg/m^2^ on days 1 and 15 and oral capecitabine 1250 mg/m^2^ in days 1–28 every four weeks) [47]. Reni et al. conducted a randomized, phase 2 monocentric study. Eighty-three patients aged 18 to 75 years with KPS of at least 70 and pathologically confirmed stage IV PDAC were randomized to receive PAXG or gemcitabine/nab-paclitaxel. At six months, 31 (74%, 95% CI 58–86) of 42 patients in the PAXG group were alive and free of disease progression compared to 19 (46%, 31–63) of 41 patients treated with the control. The mOS were 14.4 months (95% CI 2.7–37.4) and 10.7 months (95% CI 1.7–31.9) in the PAXG and gemcitabine/nab-paclitaxel group (HR 0.60—CI 95% 0.39–0.95; *p* = 0.03). The most frequent grade 3 adverse events were neutropenia (29% in the PAXG group vs. 34% in the gemcitabine/nab-paclitaxel group), anemia (21% vs. 22%) and fatigue (17% vs. 17%). The most common grade 4 adverse event was neutropenia with 12% vs. 5% in PAXG and gemcitabine/nab-paclitaxel group, respectively. Two (5%) treatment-related deaths occurred in the gemcitabine/nab-paclitaxel compared to none in the PAXG group. The achievement of the primary outcome and the relative manageability of the treatment make PAXG a valid alternative in the first line treatment of metastatic disease. A limitation of the present study is the imbalance in different basal characteristics of the patients (PS, gender, biliary stent placement, median CA19-9 concentration, and metastatic site), even if the multivariate analysis confirmed the independent prognostic value of the chemotherapy regimen. The trial was not blinded. A confirmatory, adequately powered, phase 3 study to confirm results is desirable (Table 5).

#### 5.1.4. Is FirstLine Chemotherapy Treatment with Gemcitabine Monotherapy indicated in Advanced PDAC Patients with KPS PS between 50 and 70?

In a phase III study, advanced PDAC patients were randomized to receive gemcitabine or 5FU [48]. Among enrolled patients, 87 of 126 had KPS PS between 50 and 70. In the entire population, a clinical benefit (primary endpoint) was observed in 23.8% and 4.8% of patients treated with gemcitabine and 5FU, respectively (*p* = 0.0022); median OS was prolonged (5.6 and 4.4 months in the gemcitabine and 5FU arms, respectively; *p* = 0.0025); 18-month survival was 18% in the gemcitabine arm and 2% in the 5-FU arm. The main grade 3–4 side effects of gemcitabine consisted of neutropenia (25.9%), thrombocytopenia (9.7%), nausea/vomiting (12.8%), and constipation (3.2%), without statistically significant differences as compared to 5-FU arm. The option of recommending single agent therapy in patients with KPS PS 50–70 should be considered, given the favorable risk-benefit ratio associated with gemcitabine use.

As a note of caution, it should be emphasized that the reported data refer to the entire population, including patients with KPS PS >70, so it is not clear whether the benefit and risk outcomes of the whole population are applicable to the patient subgroup considered in this recommendation (Table 5).

#### 5.1.5. Is Maintenance Treatment with Olaparib Recommended in Mutated gBRCA1–2 Metastatic PDAC, Who Are Progression-Free after at Least 4 Months of FirstLine Chemotherapy Containing a Platinum Salt?

A multicenter double-blind, placebo-controlled phase III study evaluated 154 patients (of 3315 screened patients) with gBRCA1–2 metastatic PDAC without progression of disease during at least 4 months of firstline platinum-based chemotherapy [49]. Patients were randomized to receive olaparib (92 patients) and placebo (62) at a 3:2 ratio. This study showed that mPFS was longer in the olaparib group compared with in the control group (7.4 months vs. 3.8 months, respectively; HR for disease progression or death, 0.53; 95% CI, 0.35 to 0.82; *p* = 0.004). An interim analysis of OS, at a follow-up of 46%, showed no difference between the experimental and the placebo groups (mOS: 18.9 months vs. 18.1 months, respectively; HR for death, 0.91; 95% CI, 0.56 to 1.46; *p* = 0.68). The incidence of grade 3–4 adverse events was 40% and 23% in the olaparib and control group, respectively (95% CI, −0.02 to 31). Moreover, 5% and 2% of the patients interrupted the therapy due to adverse events. PFS benefit of this therapy together with its relatively safe toxicity profile could suggest a possible role of olaparib as maintenance treatment after induction chemotherapy. However, to fully assess the benefit of this strategy, mature survival data are necessary. As of now there are no studies that can identify the population of PDAC patients that is a candidate for BRCA testing. So far, clinicians have only been able to suggest BRCA testing for PDAC patients with at least one relative with ovarian cancer or ≤50 years breast cancer, or at least two relatives with breast, pancreatic, or prostate cancer. [78] (Table 5).

#### 5.1.6. Is Surgical Resection Indicated in Patients with PDAC Oligometastatic to the Liver at Diagnosis or after Primary Chemotherapy?

Crippa et al. published the results of a bi-institutional retrospective study analyzing the role of surgical resection of 127 fit PDAC patients (ECOG PS 0–1) with liver-only metastases who underwent different primary chemotherapy protocols (gemcitabine alone or with another drug 44%; FOLFIRINOX 8%; PEXG/PDXG/PEFG 48%) [50]. Fifty-six patients (44%) had a radiological response (7%: complete response; 37%: partial response). Surgical treatment was considered for patients with complete/partial radiological response and with decrease in CA19-9 > 90% as compared with the baseline value. Of the 127 patients, 11 (8.5%) underwent surgery. The median OS was 11 months for the entire cohort and 15 months for 56 patients with complete/partial radiological response. In this subgroup survival was significantly longer in those who underwent resection (mOS: 46 months vs. 11 months; *p* < 0.0001). Independent predictors of survival were multi-agent chemotherapy (HR: 0.512), surgical resection (HR: 0.360), >5 liver metastases at diagnosis (HR: 3.515), and CA19-9 decrease < 50% compared with the baseline value (HR: 2.708). Though the study reports these interesting results, it has several limitations, such as the retrospective analysis methodology, the low number of patients who underwent surgery, the high risk of selection bias (patients with more favorable characteristics were selected for surgery).

Hackert et al. published the results of a single center retrospective study aiming at evaluating postoperative and long-term outcomes of 62 patients with PDAC of the head and oligometastatic liver disease who underwent pancreatic and liver resection [51]. The rate of clinically-relevant postoperative pancreatic fistula, postoperative hemorrhage and reoperation were 9.7%, 6.4% and 3.2%, respectively. Thirty-day postoperative mortality was 1.6%. The median OS was 12.3 months with a 5-year survival of 8.1%. Study limitations include its retrospective analysis and the availability of data regarding adjuvant therapy only for 70% of these patients (Table 5).

### 5.2. Second Line

#### 5.2.1. Is Second Line Chemotherapy Indicated in Patients with Advanced PDAC Progressing after First Line Systemic Treatment?

A phase III clinical trial randomized 46 patients with advanced PDAC and KPS PS 70–100%, progressing after first line therapy with gemcitabine, to an active treatment (OFF regimen: oxaliplatin/5FU/folinic acid; 23 patients) versus best supportive care (23 patients). The primary endpoint was OS and the original sample size was 165 patients [52]. Two systematic reviews of the literature have also considered 1503 and 3112 patients, respectively, with LAPC or metastatic PDAC progressing after first line systemic treatment and compared best supportive care arms and investigator choice (two and one studies, respectively) versus active treatment arms [53,54]. Despite early closure of the trial conducted by Pelzer at al. due to slow accrual and ethical concerns about the best supportive care arm, mOS was significantly better (4.82 months, 95% CI 4.29–5.35) in the active treatment arm (OFF regimen), as compared to the best supportive care arm (mOS 2.30 months, 95% CI 1.76–2.83), with a HR of 0.45 (95% CI 0.24–0.83, *p* = 0.008). A significant benefit in terms of median OS favoring active treatment versus best supportive care was also detected in the two systematic reviews of the literature (6 months vs. 2.8 months, *p* = 0.013, in the first study; 4.6 months vs. 2.5 months, *p* = 0.02, in the second one). In the first study, which also looked at ORR and PFS as secondary endpoints, no significant benefit in either endpoint (*p* = 0.2 and *p* = 0.26, respectively) was found in favor of active treatment.

Despite methodological limits (randomized trial of OFF versus best supportive care—small number of patients, early study closure; systematic reviews—small number of patients included in each single treatment arm; retrospective evidence derived by a pooled analysis of single treatment arms; all three studies—heterogeneous populations including patients with both LAPC and metastatic disease), results from all threeconsidered studies are consistent with the hypothesis that secondline systemic treatment might prolong survival, in patients who are fit for chemotherapy.

Along these lines, data derived from the analysis of II-line treatment(s) in the MPACT trial showed that receiving a secondline treatment (fluoropyrimidine-containing in most cases) and KPS PS ≥ 70% are among factors independently associated with longer OS at multivariate analysis, regardless of the first line treatment arm. Other factors identified in such analysis include receiving a combination of nab-paclitaxel/gemcitabine (as compared with gemcitabine monotherapy) as first line, longer PFS, and a neutrophil/lymphocyte ratio ≤5 at the end of first line [79].

In this context, taking into account the marginal impact of second line treatment on other endpoints, such as response rate and PFS, an in-depth evaluation of prognostic factors and a thoughtful selection of patients who are candidates for second line therapy appear crucial, as well as collecting and reporting QoL data, which are currently not available (Table 5).

#### 5.2.2. Is Combination Chemotherapy Indicated in Patients with Advanced PDAC Progressing afterFirst Line Systemic Treatment?

To date, no randomized trials enrolling patients progressing after first line gemcitabine/nab-paclitaxel have been reported. The NAPOLI-1 trial (randomized phase III, open label, comparative study conducted in 76 centers of 14 countries) compared a regimen of nanoliposomalirinotecan plus fluorouracil and folinic acid (nal-IRI/5FU/FA) versus a 5FUandfolinic(FF) regimen [55]. The initial study plan included a head to head comparison between nal-IRI and FF, while the combination arm was added with a subsequent amendment to the study plan. The patient population for the comparison between nal-IRI/FF and FF included 266 patients (117 and 149 in the nal-IRI/FF arm and in the FF arm, respectively) with advanced PDAC, progressing after gemcitabine-based regimens administered in neoadjuvant, adjuvant, locally advanced, or first line metastatic settings, with KPS PS ≥ 70%. The primary endpoint was to demonstrate superiority of the nal-IRI/FF versus FF regimen in terms of OS. The median OS was 6.1 months (95% CI: 4.8–8.9) for the nal-IRI/FF regimen versus 4.2 months (95% CI: 3.3–5.3) for the FF comparator (HR: 0.67, 95% CI: 0.49–0.92, *p* = 0.012), with a significant advantage in terms of PFS (HR: 0.56, 95% CI: 0.41–0.75, *p* = 0.0001) and a higher incidence of neutropenia (grade 3/4: 27% vs. 1%), diarrhea (grade 3/4: 13% vs. 4%), vomiting (grade 3/4: 11% vs. 3%), and fatigue (grade 3/4: 14% vs. 4%). Methodological limitations (heterogeneity of patient population, including subjects treated in first, second, and subsequent lines of treatment; inclusion of patients pretreated with 5FU or irinotecan; study design amended during the trial) notwithstanding, these results would support the use of nal-IRI/5-FU/FA. However, since nal-IRI is currently not reimbursed by the Italian NHS, the possible use of such regimen will not be considered further. 

The CONKO-3 trial (randomized phase III, open label, comparative study conducted at 16 German institutions) compared the OFF regimen (oxaliplatin/5-FU/FA) versus a FF regimen in a population of 165 patients with advanced PDAC progressing after first line gemcitabine monotherapy [56]. Eligible patients had to have at least one measurable target lesion and KPS PS ≥ 70%. The primary endpoint was to demonstrate OFF superiority in terms of OS. The median OS in the CONKO-3 study was 5.9 months (95% CI: 4.1–7.4) for the OFF regimen versus 3.3 months (95%CI: 2.7–4.0) for the FF comparator (HR: 0.66, 95%CI: 0.48–0.91, *p* = 0.010), with a significant advantage also in TTP (HR: 0.68, 95% CI: 0.50–0.94, *p* = 0.019) and a higher incidence of peripheral neuropathy.

In apparent contrast with the results of the CONKO-3 trial, the randomized phase III PANCREOX study did not document any advantage for the addition of oxaliplatin to a FF regimen (mFOLFOX6) in terms of ORR, PFS, and time to QoL deterioration and showed a significant detrimental effect on OS [57].

Such a discrepancy in the results of the two trials might be due to different patient selection criteria, as well as to differences in the regimens employed.

To date, no randomized phase III studies comparing the addition of irinotecan to an FF regimen versus FF alone are available. Evidence supporting the use of irinotecan-containing regimens derives from small retrospective and single-arm prospective trials (mean number of patients/individual trial—41), with PFS ranging from 2 months to 3.7 months and OS ranging from 4.2 months to 6.6 months. A randomized phase II trial comparing irinotecan/FF versus oxaliplatin/FF in a population of 61 patients pretreated with gemcitabine-based regimens found no significant differences in terms of activity and tolerability [58].

A systematic review of the literature examining 24 studies of oxaliplatin or irinotecan-containing secondline chemotherapy has not shown meaningful differences between the two approaches in terms of ORR, PFS, or OS (11.8%, 2.87 months, and 5.48 months in the entire population, respectively) [59]. However, two meta-analyses, which also included results from the NAPOLI-1 trial, suggest an advantage for irinotecan-containing regimens [60,61].

Overall, the quality of evidence supporting fluoropyrimidine-based combination regimens as second line treatment for PDAC patients progressing after first line gemcitabine/nab-paclitaxel is extremely low, due to indirectness in this specific patient population, conflicting results of the two available studies with oxaliplatin/FF regimens, and lack of randomized phase III studies evaluating nal-iri. Therefore, enrollment of patients progressing after gemcitabine/nab-paclitaxel into clinical trials should be considered upfront. If clinical trials are not a suitable option, oxaliplatin/FF or irinotecan/FF can be considered.

To date, no randomized trials enrolling patients progressing after first line FOLFIRINOX have been reported. Gemcitabine monotherapy as second line treatment in PDAC patients progressing after first line FOLFIRINOX is supported only by retrospective evidence, showing a 10% ORR, a median PFS ranging from 1.5 months to 2.5 months, and median OS ranging from 3.6 months to 5.7 months [62,63].

Small series employing the combination of gemcitabine/nab-paclitaxel as second line treatment after first line FOLFIRINOX have been reported [64]. However, no direct comparison with gemcitabine monotherapy has been reported and such combination is not authorized, nor reimbursed by the Italian NHS in this setting.

Given the extremely low quality of evidence supporting any of the available treatments in advanced PDAC patients progressing after FOLFIRINOX, enrollment into clinical trials should be considered as the main therapeutic option in this setting (Table 5).

#### 5.2.3. Are Ablative Local Treatments Useful to Improve Survival of Patients with LAPC?

Giardino et al. described a single-center experience with radiofrequency ablation (RFA) in the treatment of LAPC during laparotomy [65]. In the study period (February 2007–December 2011), 168 patients with ECOG 0–1 underwent laparotomic RFA. Of these, 107 had a follow-up > 18 months and were considered in the analysis. Forty-seven patients underwent upfront RFA followed by adjuvant chemotherapy (gemcitabine or gemcitabine plus cisplatin/oxaliplatin) or chemoradiation (group 1). Sixty patients underwent RFA after primary systemic chemotherapy or chemoradiation or intra-arterial chemotherapy (group 2). In this latter group, RFA was followed by multimodal treatment. In the entire cohort of 107 patients, postoperative morbidity and mortality rates were 25% and 1.8%, respectively. Overall mOS was 25.6 months, while patients in group 1 presented mOS of 14.7 months compared with 25.6 months in group 2 (*p* = 0.004). Thirty-two patients who underwent chemoradiation, RFA, and intra-arterial chemotherapy reached a mOS of 34 months. Study limitations included retrospective analysis, evaluation of a subset of the entire population, high heterogeneity of different treatments, patients with a more favorable tumor-biology were likely selected considering their treatments before RFA (selection bias), lack of comparison with more recent chemotherapy schedule (i.e., FOLFIRINOX, gemcitabine plus nab-paclitaxel), and with stereotaxic radiotherapy.

Martin et al. reported the results of a multicentric prospective study aimed at evaluating the efficacy of irreversible electroporation (IRE) [66]. In the study period (2010–2014), 200 patients underwent primary chemotherapy (FOLFIRINOX or gemcitabine-based) and 52% of them underwent chemoradiation as well. One hundred and fifty patients underwent IRE, while the remaining 50 patients underwent pancreatic resection plus IRE along the surgical margin if there was suspicion of incomplete resection. Overall morbidity was 37% with post-procedure mortality of 2%. Median survival was 24.9 months in the entire cohort. Median survival was 28.3 months in patients who underwent pancreatectomy and IRE versus 23.2 months for those undergoing IRE alone (*p* = ns).

Study limitations are the lack of a control group undergoing only chemotherapy/chemoradiation, heterogeneity of different treatments (chemotherapy and chemoradiation), and selection bias from considering IRE patients only without disease progression after initial chemotherapy/chemoradiation (Table 5).

#### 5.2.4. Couldthe Simultaneous Administration of Standard Oncologic Treatments Along with the Provision of Early Palliative Care Intervention in Comparison to the Standard Oncologic Treatments Alone Improve QoL or Prolong OS in Patients with Symptomatic and/or Metastatic PDAC? 

An unmet need for simultaneous care is evident in PDAC for its short life expectancy and for the impairment in QoL due to the influence of the disease-related symptoms and to the side effects associated with the treatment [67].

Maltoni et al. analyzed the role of early palliative care in Italian PDAC in two studies, reflecting the Italian reality, where patient’s awareness regarding prognosis is often limited and an unjustified confidence in the effectiveness of chemotherapy is widespread [68,69]. Two hundred and seven patients with LAPC or metastatic PDAC were randomized to receive standard antineoplastic treatment in association with early "on demand" palliative care (100 patients) or with simultaneous early palliative care (107 patients) [68]. In the experimental arm, early palliative care consisted of scheduled meetings with the palliative doctor at baseline, and then, at least every two months until death. After three months, a better QoL trend was observed in the experimental arm, even though no differences in OS, mood and family satisfaction with end-of-life care were appreciated. The impact of early systematic palliative care on health care resources and treatment aggressiveness near the end of life was also analyzed with indicators such as the use of health services available, the administration of chemotherapy in the last month of life and the coincidence of the place of death with the patient’s wishes [69]. Patients treated with parallel early palliative care showed higher usage of hospice service, a longer period of hospice care and a reduction in the administration of chemotherapy in the last 30 days of life (18.7% vs. 27.8% in the standard arm *p* = 0.036). In this group of patients, death occurred more frequently at home or in hospice (*p* = 0.102). No difference, however, was demonstrated in the frequency of admissions and access to the emergency room in comparison with the standard arm.

A pilot randomized trial conducted at the University of Pittsburgh Cancer Institute geared to assess feasibility, acceptability, and perceived effectiveness of a physician-led specialty palliative care action for patients with advanced PDAC did not achieve feasibility goals [70] (Table 5).

## 6. Follow Up

### Does the Diagnostic Anticipation of Asymptomatic Relapse through Follow-up Increase Survival of Resected PDAC?

Nordby et al. conducted a retrospective study on 164 patients who underwent pancreatectomy and a follow-up with chest–abdomen–pelvic CT every six months or at the onset of symptoms [71]. About 3/4 of the asymptomatic patients received cancer treatment compared to only 1/4 of the symptomatic patients. Furthermore, the authors observed a significant difference in terms of median time to recurrence among asymptomatic (12 months) and symptomatic (7 months) patients. Post-recurrence median survival was significantly longer in asymptomatic patients (10 months vs. 4 months, *p* < 0.00001).

Similar results were observed by MD Anderson Center researchers in a retrospective study on 327 patients [72]. Follow-up by CT identified patients with good PS and favorable biology who were more likely to benefit from specific oncology treatments. In particular, a total of 216 (66.1%) resected patients developed recurrence. Asymptomatic relapse was detected in 118 (54.6%) patients, specifically those who benefited from follow-up. The remaining patients developed symptomatic recurrence associated with multifocal disease or carcinomatosis and poor clinical status (reducing the possibility of subsequent treatments). Median recurrence time did not differ between groups, but survival after detection was shorter in symptomatic patients (5.1 months vs. 13.0 months; *p* < 0.001). Median time to recurrence was not different between groups, but median post-recurrence survival was shorter in symptomatic patients (5.1 months vs. 13.0 months, *p* < 0.001). Salvage post-recurrence treatment was more frequently administrated to asymptomatic patients than others (91.2% vs. 61.4%, *p* < 0.001). Subsequently, the authors constructed a Markov model to compare the cost-effectiveness of the following five post-surgical surveillance strategies: (1) radiological examination and investigation only at the onset of symptoms, (2–3) clinical examination and determination of Ca19-9 every three or six months, (4–5) clinical examination and abdomen–pelvis CT plus chest X-ray every threeor six months [73]. The results of this study indicated that the clinical examination and the assessment of the marker every 6 months was associated with an absolute OS of 32.8 months (compared to 24.6 months in the absence of follow-up). A more intensive follow-up strategy resulted in a worsening of cost-effectiveness without showing an actual significant clinical benefit (Table 6).

## 7. Conclusions

Though multidisciplinary management has improved survival in the resectable context, we are still far from achieving remarkable results in advanced disease. Upcoming biomarkers able to identify patients harboring a disease, yielding promising results through druggablebiotargets, are still warranted. Up to now, only biomarkers predictive for PARP inhibitor activity in gBRCA1–2 mutations or for anti-PD1 activity for MSI-high expression have proven promising, but many phase III randomized trials studies are necessary for these to become clinical practice. In conclusion, this available evidence from the AIOM panel of upper-GI experts should help clinicians in diagnosis, treatment, and follow-up. These guidelines are generally valid for one year and are updated annually, according to any clinical trials that yield innovative results that may result in changes in clinical practice. The continuous updating of the literature and the improvement of application methodologies will mean the clinician will be increasingly required to adapt to the guidelines.

## Figures and Tables

**Table 1 cancers-12-01681-t001:** Grade methodology

**a. Grade Methodology (Quality of the Evidence)**		
Quality of the Evidence		Significance
Very High		Well-performed RCTs
		Very strong evidence from unbiased observational-studies
Moderate		RCTs with some limitations
		Strong evidence from unbiased observational-studies
Low		RCTs with serious flaws
		Some evidence from observational-studies
Very Low		Unsystematic clinical observation
		Indirect evidence from observational-studies
**b. Grade Methodology (Strength of the Recommendation)**		
Strength of the Recommendation	Applicability	Significance
Strong positive	In patients with (selection criteria) the xxx intervention should be considered as a primary intention therapeutic option	The intervention in question should be considered among the first-choice therapeutic options (evidence that the benefits outweigh any harmful effects)
Weakly positive	In patients with (selection criteria) the xxx intervention can be considered as a therapeutic option	the intervention in question can be considered as a first intention option, with awareness of alternatives that can be offered (uncertainty regarding the extent to which benefits will outweigh any harmful effects). In-depth discussion with the patient is best to clarify the situation and listen to any views expressed by the patient
Weakly negative	In patients with (selection criteria), xxx intervention should not be considered as a therapeutic option	the intervention in question should not be considered as a first intention option but it may be used in highly selected cases and after full sharing of information with the patient (uncertainty regarding the extent to which harmful effects outweigh benefits)
Strong negative	In patients with (selection criteria) the xxx intervention should not be considered an option	The intervention in question must in no case be considered an option (reliable evidence that harmful effects outweigh benefits)

Abbreviations: RCTs—randomized clinical trials.

**Table 2 cancers-12-01681-t002:** Diagnosis and staging.

Quality of the Evidence	Clinical Recommendation	Strength of the Recommendation	References
Low	In patients with clinical and radiological suspicion of PDAC, pathological confirmation of diagnosis should be considered in the absence of clear signs of malignancy and in patients who are not candidates for surgery	Strong positive	[11,12]
Low	In patients with a pancreatic mass ≥ 2 cm suspected for adenocarcinoma, the execution of a contrast enhanced multislice CT should be considered the first choice for differential diagnosis and staging	Weakly positive	[13,14,15]
Low	In patients with potentially resectable pancreatic adenocarcinoma, MR could improve liver staging	Weakly positive	[16,17]
High	Multislice CT is better than MR for correct definition of non-resectability in patients with pancreatic mass, suspected for locally advanced PDAC	Weakly positive	[16,18,19,20]

Abbreviations: CT—computerized tomography; MR—magnetic resonance; PDAC—pancreatic adenocarcinoma of pancreas.

**Table 3 cancers-12-01681-t003:** Treatment of localized disease.

Quality of the Evidence	Clinical Recommendation	Strength of the Recommendation	References
High	In patients with pancreatic head tumor and jaundice, the preoperative palliation of jaundice should be avoided as it is associated with an increased risk of postoperative complications. Preoperative jaundice palliation should be limited to patients with cholangitis and/or high bilirubin levels (>15 mg/dL)	Strong negative	[21]
Low	Extended lymphadenectomy should not be considered in head PDAC with pancreaticoduodenectomy	Strong negative	[22]
Low	Pancreatic resection for resectable PDAC should be performed in high-volume or very high-volume centers to decrease postoperative morbidity and mortality	Strong positive	[23,24]
Low	In patients with resected stage I–III PDAC (R0-R1) with Karnofsky PS of at least 50%, chemoradiotherapy may be considered, with a dose of radiotherapy of at least 50 Gy (conventional fractionation and modern conformal radiotherapy techniques) in combination with gemcitabine or fluoropyrimidine	Weakly positive	[25,26,27,28,29]
High	In patients with radically resected PDAC, carefully selected with a PS ECOG 0–1 and age <70 years, an adjuvant chemotherapy treatment with mFOLFIRINOX for 6 months should be considered as the first option over to gemcitabine monotherapy	Strong positive	[30]
Low	In patients with resectable disease, perioperative treatment with PEXG could improve surgical and oncological outcomes	Weakly positive	[31,32]

Abbreviations: ECOG—Eastern Cooperative Oncology Group; Gy—Gray;mFOLFIRINOX—oxaliplatin 85 mg/m^2^, leucovorin 400 mg/m^2^, irinotecan150 mg/m^2^ and 5FU 2400 mg/m^2^c.i.; PDAC—pancreatic adenocarcinoma of pancreas; PS—performance status; PEXG—cisplatin 30 mg/m^2^, epirubicin 30 mg/m^2^, and gemcitabine 800 mg/m^2^ on days 1 and 15, every 4 weeks and capecitabine 1250 mg/m^2^ on days 1–28.

**Table 4 cancers-12-01681-t004:** Treatment of BRPC/LAPC.

Quality of the Evidence	Clinical Recommendation	Strength of the Recommendation	References
Low	In patients with BRPC preoperative treatment could increase survival in comparison with immediate surgery	Weakly positive	[33,34,35,36,37,38,39]
Very Low	Chemotherapy may be considered as a first-choice option as initial therapy, as alternative to chemoradiotherapy in patients with LAPC	Weakly positive	[40]
Moderate	In patients with LAPC who are progression-free after systemic chemotherapy (with reference to the schemes used in advanced disease), consolidation chemoradiotherapy may be considered	Weakly positive	[41,42,43]

Abbreviations–BRPC: Borderline Resectable Pancreatic cancer; LAPC: Locally Advanced Pancreatic Cancer.

**Table 5 cancers-12-01681-t005:** Treatment of advanced disease.

Quality of the Evidence	Clinical Recommendation	Strength of the Recommendation	References
High	In patients with metastatic PDAC, Karnofsky PS > 70 or ECOG PS ≥ 1, and age ≤ 70 years, a first-line chemotherapy combination of 3 or 4 drugs may be considered as a primary therapeutic option as an alternative to gemcitabine monotherapy with peculiar regard to older patients	Weakly positive	[44,45]
High	A first-line chemotherapy treatment with gemcitabine/nab-paclitaxel combination increases survival in patients with metastatic PDAC, Karnofsky PS ≥ 70 or ECOG PS ≥ 1, and age> 18 years	Strongly positive	[46]
High	In patients with metastatic pancreatic ductal adenocarcinoma, Karnofsky PS ≥ 70 or ECOG PS ≥ 1, and 18–75 years, a first-line chemotherapy combination with PAXG may be considered as a primary therapeutic option in terms of PFS and OS	Weakly positive	[47]
Very low	Weekly gemcitabine may be considered as a primary therapeutic option in patients with advanced disease and KarnofskyPS 50–70	Weakly positive	[48]
Low	Maintenance treatment with olaparib is indicated in mutated gBRCA1–2 metastatic PDAC who are progression-free after at least 4 months of first-line chemotherapy containing a platinum salt	Weakly positive	[49]
Very low	Upfront surgery is not associated with survival improvement in PDAC patients oligometastatic to the liver.	Weakly negative	[50,51]
Low	In patients affected by advanced PDAC, progressing after first-line systemic treatment and with good PS, second-line chemotherapy can be considered as the treatment of choice	Weakly positive	[52,53,54]
Very low	In patients affected by advanced PDAC, progressing after gemcitabine/nab-paclitaxel first-line systemic treatment and with good PS, second-line chemotherapy with either oxaliplatin/FF or irinotecan/FF can be considered (nal-IRI is not currently reimbursed in Italy)	Weakly positive	[55,56,57,58,59,60,61,62,63,64]
Very low	In patients with LAPC, local ablative treatments can be considered only in the context of clinical trials	Weakly negative	[65,66]
Very low	In patients with LAPC or metastatic PDAC, the combination of standard oncology care and early palliative care should be considered as a first intention option to improve quality of life and quality of care	Strong positive	[67,68,69,70]

Abbreviations: ECOG—Eastern Cooperative Oncology Group; FF—fluorine derivatives and folinic acid; LAPC—Locally Advanced Pancreatic Cancer; nal-IRI—nanoliposomalem Irinotecan; PAXG—cisplatin 30 mg/m^2^, nab-paclitaxel 150 mg/m^2^ and gemcitabine 800 mg/m^2^ on days 1 and 15 and oral capecitabine 1250 mg/m^2^ in days 1–28 every 4 weeks; PDAC—pancreatic adenocarcinoma of pancreas; PS—performance status; PFS—progression free survival; OS—overall survival.

**Table 6 cancers-12-01681-t006:** Follow-up.

Overall Quality of Evidence	Clinical Recommendation	Strength of Recommendation	References
Very low	Comprehensive follow-up of clinical examination, CEA, CA19-9, and thoracic-abdomen-pelvis CT can be considered in order to improve survival in patients with resected PDAC	Weakly positive	[71,72,73]

Abbreviations: ECOG—Eastern Cooperative Oncology Group; Gy—Gray;mFOLFIRINOX—oxaliplatin 85 mg/m^2^, leucovorin 400 mg/m^2^, irinotecan150 mg/m^2^ and 5FU 2400 mg/m^2^c.i.; PDAC—pancreatic adenocarcinoma of pancreas; PS—performance status; PEXG—cisplatin 30 mg/m^2^, epirubicin 30 mg/m^2^, and gemcitabine 800 mg/m^2^ on days 1 and 15, every 4 weeks and capecitabine 1250 mg/m^2^ on days 1–28.3. Diagnosis and Staging.

**Table 7 cancers-12-01681-t007:** Clinical findings in BRPC and LAPC.

Reference	Type of Study	Stage of Disease	Setting	Control Arm	R0 Resection Rate	OS
Versteijne et al. [34]	Meta-analysis	BRPC	NC(R)T	Surgery	88.6% vs. 63.9%, *p* < 0.0001	Median: 19.2 vs. 12.8 months
Pan et al. [35]	Meta-analysis	BRPC	NC(R)T	Surgery	OR = 4.75 [95% CI, 2.85–7.92]	HR = 0.48 [95% CI, 0.35–0.66]
Janssen et al. [36]	Meta-analysis	BRPC	NCT	NA	83.9%	Median:22.2months [95% CI, 18.8 to 25.6 months]
Chawla et al. [37]	Retrospective	BRPC	NC(R)T	Surgery	NA	Median: 25.7 vs. 19.6 months, *p* < 0.0001
Jang et al. [38]	Prospective	BRPC	NCRT	Surgery	51.8% vs. 26.1%, *p* = 0.004	Median: 21 vs. 12 months HR = 1.495 [95% CI, 0.66–3.36]
Versteijne et al. [39]	Prospective	BRPC	NCRT	Surgery	79% vs.13% *p* < 0.001	HR = 0.62 [95%CI: 0.40–0.95], *p* = 0.029
Ng et al. [40]	Meta-analysis	LAPC	CRT	CT	NA	Randomized studies: HR = 0.87 [95% CI, 0.63–1.21], *p* = 0.41 Observational studies: HR = 0.48 [95% CI, 0.35–0.60], *p* < 0.0001
Huguet et al. [41]	Systematic review	LAPC	CRT	CT	NA	HR = 0.79 [95% CI, 0.32–1.95]
Hammel et al. [42]	Prospective	LAPC	CRT	CT	NA	Median: 15.2 vs. 16.5 months; HR = 1.03 [95% CI, 0.79–1.34] *p* = 0.83
Zhong et al. [43]	Retrospective	LAPC	CRT	CT	NA	Median: 12.3 vs. 9.8 months HR = 0.79 [95% CI, 0.76–0.83] *p* < 0.001

Abbreviations: BRPC—Borderline Resectable Pancreatic cancer; CI—confidence interval; CT—chemotherapy; HR—hazard ratio; LAPC—Locally Advanced Pancreatic Cancer; NA—not achieved; NC(R)T—neoadjuvant chemo(radio)therapy; NCT—neoadjuvant chemotherapy; OS—overall survival.

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
