# Peer review of "Clinical Practice Guidelines for Diagnosis, Treatment and Follow-Up of Exocrine Pancreatic Ductal Adenocarcinoma: Evidence Evaluation and Recommendations by the Italian Association of Medical Oncology (AIOM)"

_cancers, 2020, doi:10.3390/cancers12061681_

Round 1

Reviewer 1 Report

Specific comments to the authors:

Nicola Sivestris et al. of the submitted review with the title “Clinical practice guidelines for diagnosis, treatment and follow up of exocrine pancreatic ductal adenocarcinoma: evidence evaluation and recommendations by the Italian Association of Medical Oncology (AIOM)” discuss major clinical aspects with exocrine pancreatic ductal adenocarcinoma according the SIGN and GRADE methods based on the published meta-analysis and clinical trials.

The authors span the arc from the diagnosis, treatment and to follow up of patients with exocrine pancreatic ductal adenocarcinoma. The consecutive developed evaluations and recommendations could serve as good and helpful clinical practice guidelines.

To improve and to clarify the manuscript for further submissions, the authors should include following issues in manuscript:

Overall:

  • The authors should definitively clarify, which is target group (practitioners, specialist especially oncologists).
  • Furthermore, special national features should be mentioned and discussed by the authors.
  • It is not clear, (i) how the questions were developed and (ii) how the literature was chosen and weighted by the authors for evaluation the different clinical aspects. Please describe in short.
  • All questions should be additionally gathered in a separate table including the related literature (with reference-ids) to have an good overview of all handled aspects of question exocrine pancreatic ductal adenocarcinoma.
  • Finally, the authors should add the evidence rating scale (1a-5) of the Centre for Evidence-Based Medicine, Oxford.

Details:

  • Methods: The table 1 (describing the SIGN and GRADE method) is missing, or? Please add adequately. Please correct the typo “calls among Authors” (line 66).
  • Diagnosis and Staging: Q1 - Please show and discuss separately the results of cytological and histological diagnosis. Please comment the possible role of liquid biopsies in the diagnosis algorithm. Q2 – Please include the size of the tumor lesions in the clinical recommendation.
  • Treatment of localized disease: Q5 – Please specifiy the „increased risk of postoperative complications”. Please correct the typo “than 500patiens” (line 153). Please specify the lymph node numbers of standard and extended lymphadenectomy in detail. Q6 – As the meta-analysis showed high evidence for pancreatic surgery, the data should be additionally presented in a separate figure/table to highlight the findings. Please correct the typo “resuls, Authors” (line 224). Please correct the typo “Weakl positive” (line 260-270). Please add the abbreviation of PS for Performance Status before using it the first time. Q9 - Please add descriptive recommendation although no definitive conclusions could be made according the published data, overall.
  • Treatment of BRPC/LAPC: Please add the abbreviation of BRPC and LAPC before using it the first time. As most of patients with pancreatic cancer have BRPLC or LAPC situation, the clinical findings should be presented in a separate table/figure to demonstrate the findings at a glance. Please correct the typo “VeryLow” (line 473).
  • Treatment of advanced disease: Q14 until Q16 - Why did the authors use Karnofsky instead of ECOG-performance status score. Please explain or adapt. Is the toxicity profile of the applied chemotherapy treatment with 3-4 drugs really accectable for older patients (65 to 75 years and older)? Please discuss and adjust the recommendations. Please correct the typo “VeryLow” (line 602). Q18 – Please enter the question when a BRCA1-2 testing should be performed as an additional important question for practical guidelines. Please correct the typo “all 3considered” (line 666). Please correct the typo “survivalof” (line 747).
  • Conclusions: The authors should give some recommendations or outlooks regarding upcoming biomarkers and druggable biotargets for pancreatic carcinoma as new possible diagnostic and therapeutic options in future.

Author Response

To improve and to clarify the manuscript for further submissions, the authors should include following issues in manuscript:

Overall:

  • The authors should definitively clarify, which is target group (practitioners, specialist especially oncologists).

Thanks for the comment. We clarified the target group in the text.

  • Furthermore, special national features should be mentioned and discussed by the authors.

We appreciate this advice. Accordingly, we added in the introduction these statements:

The aim of these guidelines is to standardize the multidisciplinary approach to PDAC by applying them to diagnostic and therapeutic care pathways in the regional cancer networks. All recommendations have been worked out on the basis both of up-to-date evidence from the literature and from the indications of the Italian Drug Agency (AIFA) which regulates the prescription of (antineoplastic) drugs.

  • It is not clear, (i) how the questions were developed and (ii) how the literature was chosen and weighted by the authors for evaluation the different clinical aspects. Please describe in short.

We tried to describe better the methodology and the criteria for literature search. In particular, we included the following sentences in the text:

Recommendations addressed the most relevant clinical questions investigated according to P.I.C.O (Population, Intervention, Control, Outcome) methodology. The PICO question is considered according to specific clinical features (specific characteristics of disease, stage, etc.), treatment (the therapeutic intervention in question), potential alternatives to the treatment described (describing treatments considered as alternatives to the one in question), and considering the effect of measures and of primary and secondary outcomes by summarizing the evidence, making clinical recommendations, and degrees of strength of the recommendation in tabular form. A comprehensive, exhaustive, sensitive, and reproducible bibliographic search of the sources was previously carried out on various medical-scientific databases (PubMed, Embase, CENTRAL and area-specific databases). In PubMed database, keywords were searched first through the MESH dictionary and then in "free search", using the diverse tools made available by the database with the use of search filters (age groups, type of study design, type of patients included and so on).

  • All questions should be additionally gathered in a separate table including the related literature (with reference-ids) to have an good overview of all handled aspects of question exocrine pancreatic ductal adenocarcinoma.

According with this recommendation, we gathered all the questions in separate tables (2-6) including the related literature references

  • Finally, the authors should add the evidence rating scale (1a-5) of the Centre for Evidence-Based Medicine, Oxford.  

In AIOM guidelines, the designated methodologies were PICO and GRADE. The use of the Centre for Evidence-Based Medicine (Oxford) could generate conflictual or inhomogeneous indications

Details:

  • Methods: The table 1 (describing the SIGN and GRADE method) is missing, or? Please add adequately. Please correct the typo “calls among Authors” (line 66).

We transformed all questions in GRADE and included Table 1 (a-b)

  • Diagnosis and Staging: Q1 - Please show and discuss separately the results of cytological and histological diagnosis. Please comment the possible role of liquid biopsies in the diagnosis algorithm. Q2 – Please include the size of the tumor lesions in the clinical recommendation.

In the present analysis cytological and histological approach was discussed jointly because all the major guide-lines consider these two techniques equivalent in terms of diagnostic accuracy; the choice is often related to the expertise of the single center. Although we made some modifications to the paragraph, adding also reference for the EFSUMB 2015 guide lines.

We corrected Q1 and included a brief discussion on liquid biopsies:

 In recent years, several studies have investigated the potential role of liquid biopsy in pancreatic cancer for early diagnosis. Circulating Tumor cells (CTCs), cell free DNA (cfDNA) and circulating microRNA (miRNA) can be detected in patients affected by pancreatic cancer with a rate ranging from 21% to 100% for CTCs, and with high specificity and sensitivity for miRNA panels (up to 100% and 90%, respectively) for PDAC diagnosis or high grade pancreatic intraepithelial lesions. Although the utility of liquid biopsy has been put forward, some concerns still exist regarding its extensive application in clinical practice, In the first place there is lower sensitivity and specificity compared with traditional biopsies and secondly there is a lack of consensus on the methodology for detection and assessment of CTCs, circulating cfDNA and miRNAs. Finally availability is limited only at selected laboratories and there are relatively high costs associated with such advanced technology [15].

  • Treatment of localized disease: Q5 – Please specifiy the „increased risk of postoperative complications”. Please correct the typo “than 500patiens” (line 153). Please specify the lymph node numbers of standard and extended lymphadenectomy in detail. Q6 – As the meta-analysis showed high evidence for pancreatic surgery, the data should be additionally presented in a separate figure/table to highlight the findings. Please correct the typo “resuls, Authors” (line 224). Please correct the typo “Weakl positive” (line 260-270). Please add the abbreviation of PS for Performance Status before using it the first time. Q9 - Please add descriptive recommendation although no definitive conclusions could be made according the published data, overall.

We thank the reviewer for his/her comments.

Q5 “Increased risk of postoperative complications” was modified as “higher risk of overall postoperative morbidity”

We have corrected the typo on line 153. We have added the data regarding numbers of resected lymph nodes on page 7 ( “Mean number of resected nodes ranged between 13.3 and 17.3 for standard lymphadenectomy and between 19.8 and 40.1 for extended lymphadenectomy.”.

Q6: Although the meta-analysis showed that surgery-related mortality significantly differs based on different centers experience, there are several limitations including “significant differences across the studies for the definition of high-volume center and for the characteristics of patients who underwent surgery (i.e. age and associated comorbidities), differences between healthcare systems of different countries involved in the studies (i.e. Asia, Europe, US), and the high or moderate heterogeneity for different outcomes analysis”. For these reasons, the overall quality of evidence was low.

Taking into account the limitations and the negligible relevance of this study, we would prefer to keep the data in the text only and to not provide additional Table/Figures for the different outcomes analyzed.

Q7: we have modified the typo”Weakl positive” in “Weak positive”; we have specified PS (performance status)

Q9: we added a descriptive recommendation in the text. In particular, we wrote: To date, due to the formally negative results of the primary end-point of the trial and the absence of fully published data, the use of gemcitabine plus nab-paclitaxel in the adjuvant setting is not recommended.   

  • Treatment of BRPC/LAPC: Please add the abbreviation of BRPC and LAPC before using it the first time. As most of patients with pancreatic cancer have BRPLC or LAPC situation, the clinical findings should be presented in a separate table/figure to demonstrate the findings at a glance. Please correct the typo “VeryLow” (line 473).

Thanks for the comment. We added the abbreviations, corrected the misspelling and included a new table (table 7).

  • Treatment of advanced disease: Q14 until Q16 - Why did the authors use Karnofsky instead of ECOG-performance status score. Please explain or adapt. Is the toxicity profile of the applied chemotherapy treatment with 3-4 drugs really accectable for older patients (65 to 75 years and older)? Please discuss and adjust the recommendations. Please correct the typo “VeryLow” (line 602). Q18 – Please enter the question when a BRCA1-2 testing should be performed as an additional important question for practical guidelines. Please correct the typo “all 3considered” (line 666). Please correct the typo “survivalof” (line 747).

Thanks for the comments. We correct the misspellings. Moreover, we corrected Q14-16 by inserting a clarification. In particular, we wrote: Although these studies included patients of advanced age, the sample size of the population of patients over 65-70 years of age was limited, and well-selected due to the PS cut-off, and cannot be deemed representative of elderly patients in the general population

  • Conclusions: The authors should give some recommendations or outlooks regarding upcoming biomarkers and druggable biotargets for pancreatic carcinoma as new possible diagnostic and therapeutic options in future.

We tried to improve the conclusions according to the reviewer request. In particular, we wrote: Though multidisciplinary management has improved survival in the resectable context, we are still far from achieving remarkable results in advanced disease. Upcoming biomarkers able to identify patients harboring a disease yielding promising results through druggable biotargets are still warranted. Up to now only biomarkers predictive for PARP inhibitor activity in gBRCA1-2 mutations or for anti-PD1 activity for MSI-high expression have proven promising, but many phase 3III randomized trials studies are necessary for these to become clinical practice. In conclusion, this available evidence from the AIOM panel of upper-GI experts should help clinicians in diagnosis, treatment, and follow-up. These guidelines are generally valid for one year and are updated annually according to any clinical trials that yield innovative results that may result in changes in clinical practice. The continuous updating of the literature and the improvement of application methodologies will mean the clinician will be increasingly required to adapt to the guidelines.

Reviewer 2 Report

Very informative paper.

Needs English editing extensively as it is hard sometime to follow the manuscript.

Below are some example of needed changes:

  1. Line 28-29: Need reference, if cannot add reference in abstract then maybe authors can double check to ascertain.
  2. Line 39-40: who does the incidence compare to previous years.
  3. Line 45-47: Chronic pancreatitis is associated to a 10-fold increase of PDAC risk. 46 compared to the general population while diabetes mellitus is associated to a 1.5-2 fold increase and. 47 previous gastrectomy to a 3-5 fold increase of risk, needs to be re-worded.
  4. Line 47-48: Regarding genetic factors, 10% of PDAC patients present a familiar history. Needs reference, this number seems large.
  5. Line 51-52: need the comparisons for BRCA mutations to be the same, percentages or relative risk.
  6. Line 66: authors, should not be capitalized.
  7. Line 87-89: The question needs to be clarified.
  8. Methods section needs to be re-written for clarity.
  9. Line 108: What is TC?
  10. Line 208: there are 2 numbers 5 and 3.8%?

Author Response

Comments and Suggestions for Authors

Very informative paper.

Needs English editing extensively as it is hard sometime to follow the manuscript.

Thanks for the comment. A native professional speaker revised the manuscript throughout.

Below are some example of needed changes:

  1. Line 28-29: Need reference, if cannot add reference in abstract then maybe authors can double check to ascertain.

We can not insert a reference in the abstract. Nevertheless, we included these epidemiologic data in the introduction after double-checking and added the appropriate reference there.

  1. Line 39-40: who does the incidence compare to previous years.

Thanks for the comment. We added a trend incidence.

  1. Line 45-47: Chronic pancreatitis is associated to a 10-fold increase of PDAC risk. 46 compared to the general population while diabetes mellitus is associated to a 1.5-2 fold increase and. 47 previous gastrectomy to a 3-5 fold increase of risk, needs to be re-worded.

We appreciate this comment. We reworded the sentences.

  1. Line 47-48: Regarding genetic factors, 10% of PDAC patients present a familiar history. Needs reference, this number seems large.

We corrected the incidence according to the reference reported.

  1. Line 51-52: need the comparisons for BRCA mutations to be the same, percentages or relative risk.

Thanks for the comment. We corrected it.

  1. Line 66: authors, should not be capitalized.

Thanks for the comment. We corrected it.

  1. Line 87-89: The question needs to be clarified.

We corrected Q1.

  1. Methods section needs to be re-written for clarity.

We tried to better describe the methodology.

  1. Line 108: What is TC?

Thanks for the comment. We corrected it.

  1. Line 208: there are 2 numbers 5 and 3.8%?

Thanks for the comment. We corrected it.

Round 2

Reviewer 2 Report

A very well written and informative paper. We can accept as is, however, the changes below are needed:

  1. Line 197, page 5, Korea and not Corea.
  2. Discussions on questions 1-5 have no conclusion, we recommend authors providing their conclusions or recommendations based on the data.
  3. Line 234 "and 5 e 3.8%", I am not sure what this is but needs to be corrected.
  4. Q13 needs conclusion/recommendation from the panel.

Author Response

We corrected the mistakes (Corea and e with Korea and and, respectively).

We added the conclusions required.